# Comparative Study of the Proteins Involved in the Fermentation-Derived Compounds in Two Strains of *Saccharomyces cerevisiae* during Sparkling Wine Second Fermentation

**DOI:** 10.3390/microorganisms8081209

**Published:** 2020-08-08

**Authors:** María del Carmen González-Jiménez, Teresa García-Martínez, Juan Carlos Mauricio, Irene Sánchez-León, Anna Puig-Pujol, Juan Moreno, Jaime Moreno-García

**Affiliations:** 1Department of Agricultural Chemistry, Edaphology and Microbiology, Microbiology Area, Agrifood Campus of International Excellence ceiA3, University of Cordoba, 14014 Cordoba, Spain; b02gojim@uco.es (M.d.C.G.-J.); mi2gamam@uco.es (T.G.-M.); b32salei@uco.es (I.S.-L.); qe1movij@uco.es (J.M.); b62mogaj@uco.es (J.M.-G.); 2Department of Enological Research, Institute of Agrifood Research and Technology-Catalan Institute of Vine and wine (IRTA-INCAVI), 08720 Barcelona, Spain; anna.puig@irta.cat

**Keywords:** sparkling wine, second fermentation, fermentation by-products, *Saccharomyces cerevisiae* flor yeast, proteins

## Abstract

Sparkling wine is a distinctive wine. *Saccharomyces cerevisiae* flor yeasts is innovative and ideal for the sparkling wine industry due to the yeasts’ resistance to high ethanol concentrations, surface adhesion properties that ease wine clarification, and the ability to provide a characteristic volatilome and odorant profile. The objective of this work is to study the proteins in a flor yeast and a conventional yeast that are responsible for the production of the volatile compounds released during sparkling wine elaboration. The proteins were identified using the OFFGEL fractionator and LTQ Orbitrap. We identified 50 and 43 proteins in the flor yeast and the conventional yeast, respectively. Proteomic profiles did not show remarkable differences between strains except for Adh1p, Fba1p, Tdh1p, Tdh2p, Tdh3p, and Pgk1p, which showed higher concentrations in the flor yeast versus the conventional yeast. The higher concentration of these proteins could explain the fuller body in less alcoholic wines obtained when using flor yeasts. The data presented here can be thought of as a proteomic map for either flor or conventional yeasts which can be useful to understand how these strains metabolize the sugars and release pleasant volatiles under sparkling wine elaboration conditions.

## 1. Introduction

Sparkling wine is a very distinctive wine with a unique winemaking process. Its peculiarity is mainly due to a second fermentation performed in closed bottles, where wines acquire an effervescent characteristic. This is followed by a long aging process, in which the wine is in contact with the yeast lees and thereby affecting its organoleptic properties. Its production, despite being lower compared to that of still wines, has an extensive economic impact on the enology industry. This is due to the relatively high economic value of most sparkling wines [1].

Sparkling wine elaboration by the *champenoise* or traditional method (like champagne in France and cava in Spain) involves two main steps. First, a fermentation where the grape must be converted to wine and second, a process called “*prise de mousse*” [2]. The latter consists of a secondary fermentation process in sealed bottles after adding sugar and yeast, followed by at least nine months of aging on lees at low temperature (12–16 °C). During the “*prise de mousse*”, yeasts are subjected to several stress factors, such as high ethanol content, nitrogen deficiency, low pH values, low temperature and CO_2_ overpressure [3]. These affect yeast metabolism and contribute to important modifications of sparkling wine organoleptic properties [4]. During fermentation, the yeast produces ethanol and carbon dioxide, among others, which, despite being toxic, the yeasts are able to cope with. It is during the aging of the wine in contact with the lees when mannoproteins are released as well as compounds derived from autolysis and enzymes involved in reactions that affect some aroma precursors [5,6].

A large number of studies have reported the metabolic/enzymatic potential of certain non-conventional yeasts and their role in improving some technological and sensory aspects of wine [7,8,9,10], such as the positive effect on aroma, glycerol, polysaccharides, mannoproteins, and volatile acid [11,12]. Therefore, the use of non-conventional yeasts in wine fermentations has become a current trend in the wine industry. These unconventional yeast species are used in winemaking with objectives such as (i) control the acidity [11], (ii) improve color extraction and mouthfeel [13], (iii) reduce the ethanol content [8,14]; and, more recently, and (iv) improve foam properties in sparkling wines [15].

Due to the capacity of this yeast to support high concentrations of ethanol, the use of a non-conventional yeast, such as *Saccharomyces cerevisiae* flor yeast strains for sparkling wine elaboration is suitable and is a possible advantage for the industry of wine. Further, flor yeast strains possess distinctive characteristics compared to other fermentative *S. cerevisiae* strains, such as their capacity to form a biofilm on the air-liquid interface of the wine for the elaboration of Sherry wines [16]. These cell adhesive properties allow the winemakers to easily remove yeasts and sediment in the “degüelle” phase during the production of cava by the traditional method. Moreover, previous studies have demonstrated that flor yeast is a good candidate for sparkling wine elaboration because they produce volatile compounds, such as higher alcohols, aldehydes, esters and ketones, and its influence on the wine final aroma [17,18]. The use of flor yeast could reduce production cost and time, marking a significant step forward in the sparkling wine industry. At the same time, it mitigates the current situation of low diversity of commercially available yeasts for winemaking, in this particular case for the production of sparkling wines by the traditional method.

Here, we aim to reveal the yeast proteins responsible for the production of fermentation compounds released in the second fermentation during the elaboration of cava by a flor *S. cerevisiae*, and compare it with a conventional strain used in the production of this type of wine. The data presented can be seen as a proteomic map of either the flor or conventional yeasts which can be useful to understand how these strains metabolize the sugars and release pleasant volatiles under *prise de mousse* conditions. This knowledge can shed light on the molecular mechanism behind the production of characteristic volatile compounds that will determine the odorant profiles of this type of wine.

## 2. Materials and Methods

### 2.1. Microorganism and Experimental Conditions

The microorganisms used were two strains of *S. cerevisiae*. The first strain, *S. cerevisiae* G1 (ATCC: MYA-2451), is an industrial flor wine yeast from the collection of the Department of Microbiology of the University of Cordoba (Spain). It was isolated from Fine Sherry wine of Montilla-Moriles designation of origin (DO) (Spain). This strain forms a thick biofilm (velum) about 30 days after inoculation with a cell viability higher than 90% [19]. The second strain, *S. cerevisiae* P29 (CECT 11770), was used as the control strain and isolated in the Penedès grape-growing area (Spain) by the Catalan Institute of Vines and Wines (INCAVI). INCAVI recommends P29 for the elaboration of “cava” Spanish sparkling wine. A standardized commercial base wine, obtained by fermenting musts from Macabeo and Chardonnay grapes in a proportion 6:4, was used for the second fermentation. After settling, the base wine was subjected to a second fermentation inside 750 mL bottles at 14 °C. Sucrose and yeast cells were added to the base wine to reach 22 g/L per bottle and 1.5 × 10^6^ cells/mL.

The changes caused by yeast during the second fermentation were monitored at three sampling times: (i) the base wine (T0) (ii) at the middle of fermentation stage, when CO_2_ pressure reached 3 bar (MF); and (iii) at the end of the second fermentation (EF) one month after when CO_2_ pressure reached 6.5 bar. Data shown in Figure 1. All the samples were analyzed in triplicate.

### 2.2. Proteome Analysis

The cells were collected from each bottle by centrifugation at 4500× *g* for 10 min by a centrifuge (Hettich^®^ ROTINA 38/38R, Kirchlengern, Germany) and the sediment was washed twice with sterile distilled cold water. Afterwards, cells were broken by a mechanical technique in Vibrogen Cell Mill V6 (Edmund Bühler, Bodelshausen, Germany) using 500 µm diameter glass balls (Sigma-Aldrich, Darmstadt, Germany). Once the cells were broken, the protein pull was extracted. For this purpose, extraction buffer and protease inhibitors cocktails were used. A total of 500 µg of protein of each condition and replica was loaded. The OFFGEL high-resolution kit, pH 3–10 (Agilent Technologies, Palo Alto, CA, USA) was used for protein preparative isoelectric focusing (IEF) in solution. Protein samples were solubilized in protein OFFGEL fractionation buffer (Agilent Technologies, Part number 5188–6444, Santa Clara, CA, USA), and aliquots evenly distributed in 12-well 3100 OFFGEL fractionator trays according to the supplier’s instructions. Proteins from each well were scanned and fragmented on an LTQ Orbitrap XL mass spectrometer (Thermo Fisher Scientific, San Jose, CA, USA) equipped with a nano LC Ultimate 3000 system (Dionex, Germering, Germany). To obtain the concentration of a protein in the sample, Exponentially Modified Protein Abundance Index (emPAI) was used [21]. These procedures and methods are described in more detail by Moreno-Garcia et al. (2015) [22] and Porras-Agüera et al. (2020) [23].

The quantified aroma compounds were related to proteins directly involved in their metabolism using the following databases: YMDB (yeast metabolome database; http://www.ymdb.ca/), SGD (*Saccharomyces* genome database; http://www.yeastgenome.org/), and Uniprot (http://www.uniprot.org/).

### 2.3. Statistical Analysis

The software package Statgraphics Centurion XVI.II, (STSC, Inc., Rockville, MD, USA) was used for statistical analysis of the proteins. A multiple-sample comparison procedure (MSC) was used to compare two or more independent samples via ANOVA and Fisher’s test to establish homogenous groups at a level of significance of 95% (*p*-value < 0.05). Data were previously normalized according to root square and Pareto scaling, to avoid the differences introduced by the measurement units [24]. All treatments were evaluated in triplicate.

In addition, a correlation analysis to establish significant relationships between metabolites and proteins were carried out according to Metaboanalyst (https://www.metaboanalyst.ca/).

## 3. Results and Discussion

A total of 50 proteins and 43 proteins related to the metabolism of fermentation metabolites (ethanol, glycerol, acetic acid, acetaldehyde, acetoin, and 2,3-butanediol) have been identified in flor yeast and conventional yeast, respectively (Table 1, Appendix A). The fermentation related proteins have been sorted in subpathways. Each subpathway is commented and discussed separately.

### 3.1. Glycolysis/Gluconeogenesis Proteome

A total of 25 and 23 proteins involved with the glycolysis/gluconeogenesis pathway out of a total of 38 proteins currently documented in *S. cerevisiae*, were identified in the flor yeast and conventional strain, respectively. The contents of the different proteins were analyzed at different times in the second fermentation in the production of sparkling wine (T0, MF, and EF).

In general, the proteomic profiles obtained in both strains for the proteins involved in these pathways were not remarkably different. Content of proteins like glyceraldehyde-3-phosphate dehydrogenases (Tdh1p, Tdh2p and Tdh3p), Pgk1p and enolases (Eno1p and Eno2p), progressively increased during the second fermentation in both strains but the increase was more abrupt in the flor yeast (Appendix A). These proteins catalyze the reversible steps 1, 2, and 4 of the subpathway that synthesizes pyruvate from D-glyceraldehyde 3-phosphate, steps shared by glycolysis and gluconeogenesis pathways. An increased synthesis of glyceraldehyde-3-phosphate dehydrogenases during the second fermentation could be related to the recycling of NAD^+^/NADH for the continuation of glycolysis; otherwise, the glycolytic flow would decrease, which could lead to exhaustion of the ATP energy charge, making it lethal for yeast [26]. Most of the NADH produced during glycolysis is used by yeast for the formation of ethanol from acetaldehyde. Figure 2 and Figure 3 prove a significant inverse correlation between glycolysis/gluconeogenesis protein content and glucose concentration, indicating that enzymes are degrading glucose. However, at EF, when fermentable carbon sources are depleted (~0.3 g/L) and the major carbon sources are ethanol or glycerol, yeast Tdhps, Pgk1p and Enops can catalyze for gluconeogenesis [16,22]. Recently, Porras-Agüera et al. (2019) postulated that the increase in Tdhps content could be related to cell death or stress response, and thus proposing them as possible cell death biomarkers during the second fermentation [20]. A higher abundance of gluconeogenesis-related proteins in the flor yeast versus the conventional sparkling yeast may be related to an evolutionary adaptation of the first strain to media with high concentrations of non-carbon sources where flor yeasts are predominant [27].

Cell wall proteins glucanases Exg1p, Exg2p, and Bgl2p that hydrolyze β-glucan chains in the cell wall leading to the release of glucose, were also reported in higher concentrations in the flor yeast. The presence of these proteins at early stages of the fermentation may be related to cell expansion during growth while at the end of fermentation can be involved in cell wall degradation [5]. One more protein that reported an increase in its concentration at EF in flor yeast is Suc2p, which was found 5-fold higher than in the conventional strain. This protein is capable of transforming sucrose into glucose and fructose. Its non-glycosylated form is expressed constitutively, while its glycosylated form is regulated by the repression of glucose [28]. The absence of glucose in the medium at the end of the second fermentation causes the yeasts to synthesize the non-glycosylated form. This enzyme is excreted into the periplasmic space, where the hydrolysis occurs, and letting the monosaccharide products of the reaction, glucose and fructose, be transported into the cell. Higher Suc2p contents at EF in the flor yeast may be associated to a higher cell wall degradation that correlates with higher autolysis proteins compared to the conventional strain [18,23].

### 3.2. Proteins Related to the Metabolism of Pyruvate to Ethanol and Acetic Acid

A total of 16 and 12 proteins involved in the formation of ethanol from pyruvate (out of a total of 22 proteins currently documented in *S. cerevisiae*) were identified in the flor yeast and conventional strain, respectively (Appendix A).

Adh1p stood out by ranging in content from 0.4 to 1.8 (mol%) with a positive trend towards EF. Adh1p and Adh5p contents were reported two-fold more in concentration for the flor yeast compared to the conventional strain in MF and EF; where EF > MF. These enzymes are responsible for reversible exchange of acetaldehyde and ethanol during glucose fermentation. This increase is related to the drastic decrease in the amount of acetaldehyde quantified at the end of the second fermentation and the increase in the amount of ethanol produced, indicating a reaction direction towards ethanol (Figure 2 and Figure 3). At concentrations below 70 mg/L, acetaldehyde can provide a fruity flavor to wine, which often occurs in freshly fermented wines. However, it can be pungent and negative over 100 mg/L and is often associated with bruised apple, Sherry, walnut and oxidation [29]. In this study, both the flor yeast and the conventional yeast presented concentrations lower than 100 mg/L at the end of the second fermentation [17,25] supporting the use of flor yeast in sparkling wine second fermentation. Furthermore, pyruvate decarboxylases Pdc1p and Pdc5p highlighted in MF for high protein content in flor yeast compared to the conventional strain. Both pyruvate decarboxylases are key enzymes in alcoholic fermentation and are responsible for the decarboxylation of pyruvate to acetaldehyde. Pdc1p is the main active form during glucose catabolism, while Pdc5p is a secondary form that is only expressed under thiamine starvation [30]. This higher protein content was reflected in an increased acetaldehyde concentration in the middle of the second fermentation in flor yeast (Figure 2). High concentration of Adhps in flor yeast at EF, when no sugars remain, may be related to an adaptive proteomic response of these yeasts to environments with only non-fermentable carbon sources [16,31]. However, the anoxia inside the bottles does not allow ethanol consumption so the alcohol dehydrogenases, although abundant, may remain inactive.

The metabolism of acetic acid in *S. cerevisiae* is primarily synthesized as an intermediate by cytosolic pyruvate dehydrogenase bypass. This involves the conversion of pyruvate to acetaldehyde by pyruvate decarboxylase, which is subsequently oxidized to acetate by the action of aldehyde dehydrogenase (*ALD*) [32,33,34]. This acetic acid is key for the formation of fatty acids, acetyl-CoA, through the action of acetyl-CoA synthetase (*ACS*), so there must be a balance between *ALD* and *ACS* activity. The optimal concentration in wine is less than 0.20 g/L [35]. In excessive amounts, acetic acid gives wine a pungent taste and an unpleasant aroma of vinegar [36].

A total of seven proteins related to the acetic acid metabolism were identified in both strains, out of a total of 14 documented proteins in *S. cerevisiae*. However, in quantitative terms, the proteomic profile obtained for each strain was different (Appendix A). At T0, Ald4p stands out for higher protein content in conventional strain versus flor yeast, however in MF, this protein was two-fold more in protein content in flor yeast. As mentioned above, Ald4p together with Ald5p, participate in a pathway of mitochondrial pyruvate dehydrogenase, in which pyruvate is first decarboxylated to acetaldehyde in the cytosol by pyruvate decarboxylase and then converted to acetate by mitochondrial acetaldehyde dehydrogenases [37]. In general, the protein content of Acs2p reported was lower compared to Ald4p. According to Verduyn et al. (1990) [38] less acetic acid is produced if *ALD* activity is lower than *ACS* activity. This can explain the significant increase in acetic acid during the course of the second fermentation in flor yeast [17].

Even though no protein was identified at the end of the second fermentation in flor yeast and almost not quantifiable in the conventional yeast, the amount of acetic acid produced was higher at this sampling time. This fact could be correlated to the decrease in the amount of acetaldehyde, previously described [17,25]. This conversion could have taken place when the cells are still performing the alcoholic fermentation when the proteins were detectable, and the acetic acid remains until the sampling time.

### 3.3. Proteins Related to the Metabolism of Pyruvate-Acetoin-2,3-Butanediol

Acetoin and 2,3-butanediol are by-products generated by *S. cerevisiae* during alcoholic fermentation that confer buttery and cream aromas, when present over concentrations of 0.03 and 0.67 g/L, respectively. Acetoin can also be a precursor of some off-odor compounds, such as diacetyl. High fermentative wine yeasts generally produce low acetoin levels [39]. In this study, the conventional strain produced less acetoin than the flor yeast, however, the differences were not significant. The acetoin/2,3-butanediol pathway in yeasts contributes to detoxification of acetaldehyde because acetoin is a weaker inhibitor than acetaldehyde [40]. Out of the five proteins documented in *S. cerevisiae* (Bdh1p, Bdh2p, Pdc1p, Pdc5p, and Pdc6p) involved in the formation of 2,3-butanediol, only two were identified (Pdc1p and Pdc5p) in both strains (Appendix A). In MF, higher contents were observed for Pdc1p and Pdc5p in flor yeast. In both cases a direct correlation could be established between the quantity of acetoin quantified and Pdc1p and Pdc5p, and this correlation was stronger for flor yeast except for the conventional yeast Pdc5p (Figure 2 and Figure 3).

On the other hand, Bdh1p and Bdh2p were not reported in any of the strains involved in the reversible oxidation of acetoin to 2,3-butanediol and the irreversible reduction of 2,3-butanediol to (S)-acetoin, respectively. 2,3-Butanediol represents an important source of aroma [41] although it has a very high odor threshold value (~150 mg/L). In wine, its concentration varies from approximately 0.2 to 3 g/L, with an average value of approximately 0.57 g/L. This high content can have some effect on the wine bouquet due to its slightly bitter taste and also on the body of the wine due to its viscosity [41]. The changes in 2,3-butanediol and the absence of Bdh1p and Bdh2p could be attributed to limitation in the detection method that could not quantify very low protein content or to a potential activity of another enzyme that catalyzes the same reaction or another reaction known to produce this compound.

### 3.4. Proteins Related to the Metabolism of Dihydroxyacetone Phosphate-Glycerol.

Glycerol is quantitatively the most important fermentation product after ethanol and carbon dioxide, its concentration depends on environmental factors, such as temperature, aeration, sulfite level, and yeast strain [42]. Glycerol contributes positively to the sensory quality of the wine, providing smoothness and viscosity [43]. In *S. cerevisiae*, this polyol plays two main roles in physiological processes: it fights osmotic stress and controls intracellular redox balance, and [44,45,46] converts excess NADH generated during biomass formation to NAD^+^. Glycerol is synthesized by reducing dihydroxyacetone phosphate to glycerol 3-phosphate and is catalyzed by an NAD-dependent cytosolic G3P dehydrogenase (*GPD*), followed by dephosphorylation of glycerol 3-phosphate by a specific phosphatase (*GPP*).

In this work, 11 proteins have been identified (out of 19 documented proteins in *S. cerevisiae*) related to glycerol metabolism in both strains (Appendix A). In general, the proteomic profile obtained for each sampling time was similar in both strains, but a higher protein content was reported in the case of flor yeast. Gpd1p, Gpp1p, and Gpp2p almost doubled in quantity in T0 and MF in the conventional yeast versus flor yeast. Remize et al. (2003) detected that Gpd1p increases during the growth phase. The beginning of the second fermentation in sparkling wine involves anaerobiosis and osmotic stress that influence the expression of *GPD* genes [47]. Under these conditions, the respiratory chain does not function and the production of glycerol is the only possible mechanism of re-oxidation of NADH. Gpd1p, Gpd2p (not identified), Gpp1p and Gpp2p play a major role in glycerol formation. Depending on the way in which they are combined, they have been related to the production of glycerol during osmotic stress (Gpd1p-Gpp2p combination) or to the adjustment of the NADH-NAD^+^ redox balance under anaerobic conditions (Gpd2p-Gpp1p combination) [47]. In this work, the first combination (Gpd1p-Gpp2p) has been reported for both yeast strains. Both yeasts increased the synthesis of both proteins at T0 and MF which would promote the accumulation of glycerol inside the cell to withstand the osmotic stress. However, in flor yeast these two proteins were not identified at EF. Further, no significant differences were obtained in the extracellular glycerol concentration in flor yeast while in conventional yeast there was a significant decrease in concentration from MF to EF [17,25]. It was not possible to establish any significant correlation between the concentration of this metabolite and the content of the proteins involved in any of the strains (Figure 2 and Figure 3). Also, glyceraldehyde-3-phosphate dehydrogenases, previously commented, can influence the glycerol concentration (maybe producing at MF and consuming at EF). These results suggest that there is a prevalence of the metabolic pathway of ethanol production versus that of glycerol formation since considerable increase in the ethanol concentration was obtained while the glycerol concentration remained stable, possibly due to an accumulation of this compound inside the cell. Another possible explanation for this fact is that these proteins have not been activated by yeast, causing a change in the coenzyme requirement during the synthesis of glutamate from NADPH to NADH, and decreasing the availability of NADH for the synthesis of glycerol and an increase in the yield of ethanol [48]. The balance between the concentration of ethanol and glycerol results in pleasant and stable wines.

The metabolites and proteins that displayed the highest concentration in each strain are highlighted in a schematic figure (Figure 4 and Figure 5) to provide a better understanding of the results obtained in this work.

## 4. Conclusions

This work is focused on the use of an unconventional flor yeast to produce sparkling wine by comparing with a typical wine yeast strain. The relationship of the yeast proteome with the exo-metabolites excreted in the medium during the second fermentation in the production of sparkling wine has been established.

Fifty proteins and 43 proteins related to the metabolism and transport of fermentation metabolites (ethanol, glycerol, acetic acid, acetaldehyde, acetoin, and 2,3-butanediol) have been identified in flor yeast and conventional yeast, respectively. Not remarkable differences were found among the tested strains, but a lower concentration of most proteins was reported in the conventional yeast. Consequently, the concentration of the related metabolites was different in each strain and all above their odor threshold.

This study highlights that flor yeasts generally used to produce Sherry wine, can perform autolysis at high levels during the second fermentation and improve the quality and diversity of sparkling wine. In view of the results obtained, the use of this type of flor yeast is suggested for the production of sparkling wine. In addition, this type of yeast can resist high content of ethanol and has high adhesion properties. These characteristics make it an ideal candidate for the production of sparkling wines.

## Figures and Tables

**Figure 1 microorganisms-08-01209-f001:**
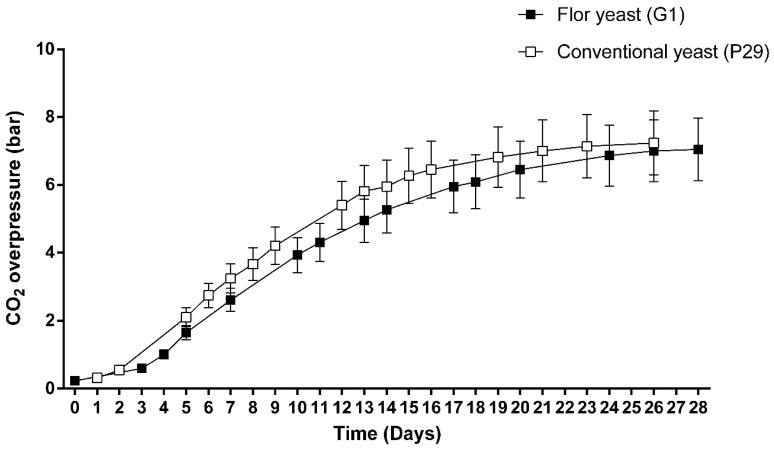
Evolution of endogenous CO_2_ released by flor yeast and conventional yeast during the second fermentation in Spanish sparkling wine (cava) elaboration (Porras-Agüera et al., 2019) [20].

**Figure 2 microorganisms-08-01209-f002:**
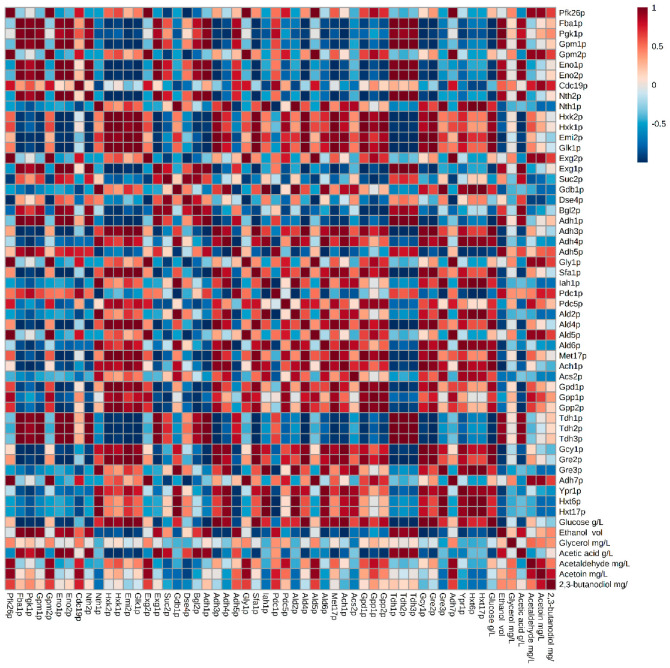
Matrix of correlations established between proteins and compounds released in the second fermentation in the flor yeast G1 strain. Metabolome data extracted from Martínez-García et al. (2020) [17].

**Figure 3 microorganisms-08-01209-f003:**
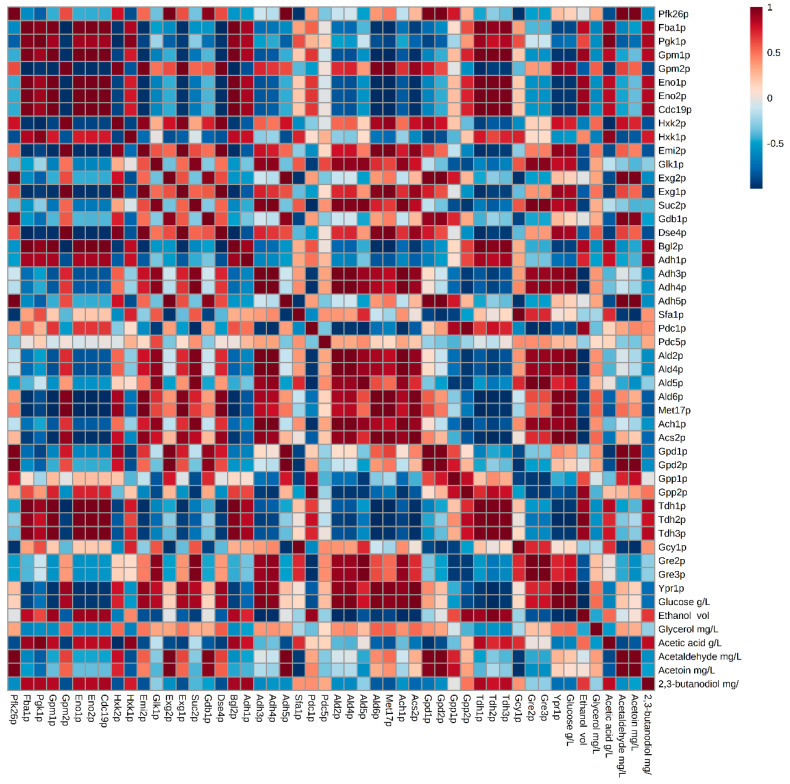
Matrix of correlations established between proteins and compounds released in the second fermentation in the conventional yeast P29 strain. Metabolome data extracted from Martínez-García et al. (2017) [25].

**Figure 4 microorganisms-08-01209-f004:**
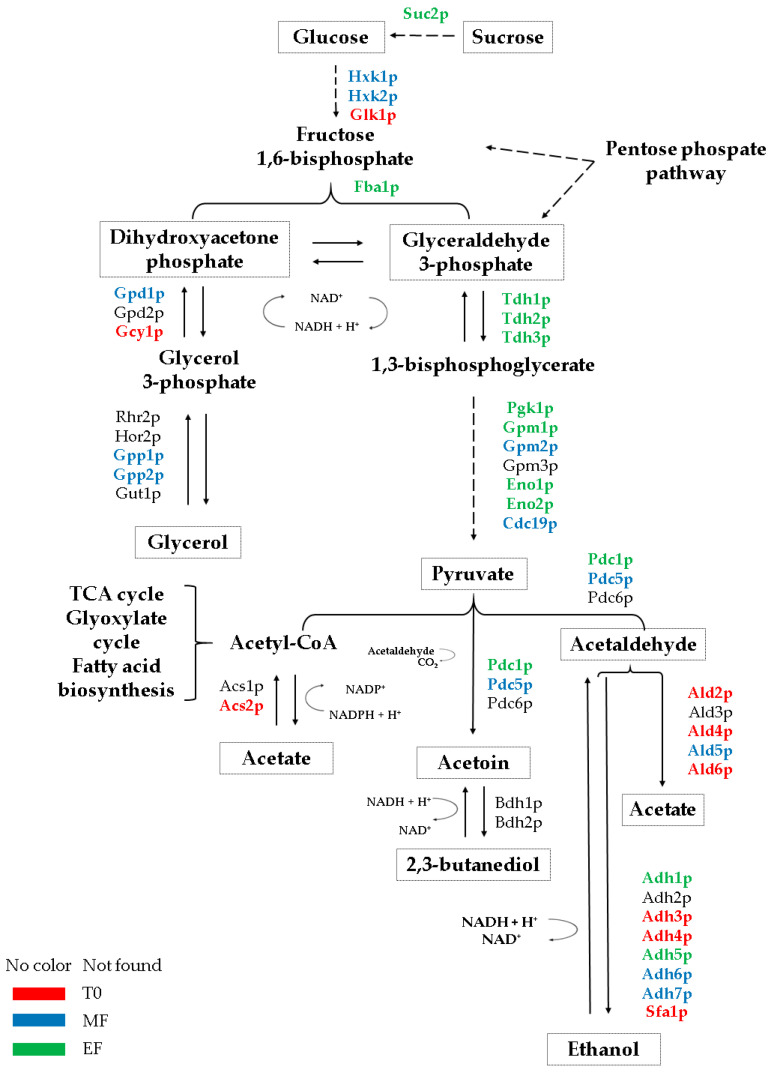
Summary of the scheme of proteins involved in the compounds derived from *Saccharomyces cerevisiae* flor yeast fermentation during the second fermentation in the production of sparkling wine. The color of the protein names represents the condition in which the highest protein content of the proteins was identified. Each condition is represented by a color: red for the base wine, T0; blue for the middle of the fermentation, MF; green for the end of the second fermentation, EF.

**Figure 5 microorganisms-08-01209-f005:**
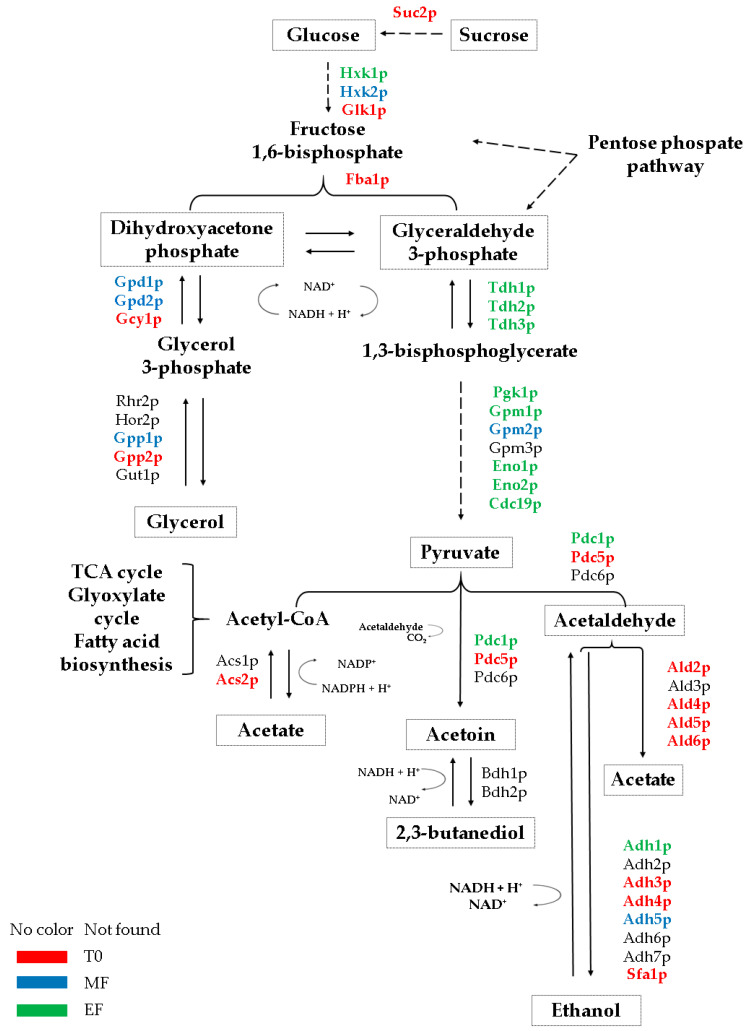
Summary of the scheme of proteins involved in the compounds derived from fermentation in *Saccharomyces cerevisiae* conventional yeast during the second fermentation in the production of sparkling wine. The color of the protein names represents the condition in which the highest protein content of the proteins was identified. Each condition is represented by a color: red for the base wine, T0; blue for the middle of the fermentation, MF; green for the end of the second fermentation, EF.

**Table 1 microorganisms-08-01209-t001:** Composition of the base wine (T0) and the Spanish sparkling wine at the middle of the second fermentation (MF) and at the end of the second fermentation (EF) in flor yeast and conventional yeast strains. Data provided by Martínez-García et al. (2017) and (2020) [17,25].

	Flor Yeast			Conventional Yeast		
	T0	MF	EF	T0	MF	EF
**Ethanol** (**% *v*/*v***)	10.23 ± 0.02	10.76 ± 0.04	11.4 ± 0.1	10.23 ± 0.02	10.85 ± 0.04	11.60 ± 0.03
**Acetaldehyde** (**mg/L**)	87 ± 1	132 ± 1	87 ± 16	87.2 ± 1.1	133.9 ± 7.3	85.2 ± 0.2
**Acetoin** (**mg/L**)	19 ± 1	61 ± 1	31 ± 2	19.3 ± 1.3	127.5 ± 11.8	24.5 ± 6.3
**2,3-butanediol** (**mg/L**)	171 ± 7	221 ± 4	200 ± 33	171 ± 6	166 ± 4	192 ± 12
**Acetic acid** (**g/L**)	0.23 ± 0.02	0.20 ± 0.00	0.28 ± 0.02	0.23 ± 0.02	0.20 ± 0.00	0.22 ± 0.00
**Glycerol** (**mg/L**)	4020 ± 656	4493 ± 164	4227 ± 297	4019 ± 655	4557 ± 212	3513 ± 163

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
