# Peer review of "Comparative Study of the Proteins Involved in the Fermentation-Derived Compounds in Two Strains of Saccharomyces cerevisiae during Sparkling Wine Second Fermentation"

_microorganisms, 2020, doi:10.3390/microorganisms8081209_

Round 1

Reviewer 1 Report

Investigation of yeasts proteins responsible for the production of fermentation compounds during the second fermentation is of great importance in winemaking industry. In this study, the differences between a flor and conventional Saccharomyces strains were investigated. The results are of great interest and provide new insights regarding the molecular mechanism behind the formation of volatile compounds. Paper is well written and results and discussion are appropriate.

Few suggestions/critical advices are reported below:

-provide a Graph with the CO2 pressure evolution during second fermentation.

-provide a Table with the chemical composition of the wines at the end of the monitored period.

Author Response

We appreciated very much the reviewer´ comments.

A file is attached and all the changes made in the original manuscript can be seen in the revised version (it has been saved in blue).

Reviewer 2 Report

The paper is interesting but needs an extensive editing of English language and style. In this form, it is harsh to understand and it is not readably.

Moreover, I have some doubt regarding the use of only two strains.

Author Response

We appreciated very much the reviewer´ comments.

A file is attached.

All the changes made in the original manuscript can be seen in the revised version (it has been saved in blue).

Round 2

Reviewer 2 Report

OK